**Climatology of mesopause region nocturnal temperature, zonal wind, and sodium density observed by sodium lidar over Hefei, China (32°N, 117°E)**

Tao Li[1*], Chao Ban[1,2*], Xin Fang[1], Jing Li[1], Zhaopeng Wu[1], Wuhu Feng[3,4], John M. C. Plane[3], Jianguang Xiong[5], Daniel R. Marsh[6], Michael J. Mills[6], and Xiankang Dou[1]

[1]CAS Key Laboratory of Geospace Environment, School of Earth and Space Sciences, University of Science and Technology of China, Hefei, Anhui, China

[2]Now at Institute of Atmospheric Physics, Chinese Academy of Sciences, Beijing, China[3]School of Chemistry, University of Leeds, Leeds, United Kingdom

[4]NCAS, School of Earth and Environment, University of Leeds, Leeds, United Kingdom

[5]Institute of Geology and Geophysics, Chinese Academy of Sciences, Beijing, China

[6]National Center for Atmospheric Research, Boulder, CO, USA

*To whom correspondence should be addressed: litao@ustc.edu.cn; banchao@mail.iap.ac.cn*

**Abstract**

The University of Science and Technology of China narrowband sodium temperature/wind lidar, located in Hefei, China (32°N, 117°E), has made routine nighttime measurements since January 2012. 154 nights (~1400 hours) of vertical profiles of temperature, sodium density, and zonal wind, and 83 nights (~800 hours) of vertical flux of gravity wave (GW) zonal momentum in the mesopause region (80-105 km) have been obtained during the period from 2012 to 2016. In temperature, it is most likely that the diurnal tide dominates below 100 km in spring, while the semidiurnal tide dominates above 100 km throughout the year. A clear semiannual variation in temperature is revealed near 90 km, in phase with the tropical mesospheric semiannual oscillation (MSAO). The variability of sodium density is positively correlated with temperature below 95 km, suggesting that in addition to dynamics, the chemistry also plays an important role in the formation of sodium atoms. The seasonal variability of sodium density observed by both lidar and satellite generally agrees well with a whole atmosphere model simulation using an updated meteoric input function which includes different cosmic dust sources. In zonal wind, the diurnal tide dominates in both spring and fall, while semidiurnal tide dominates in winter. The observed semiannual variation in zonal wind near 90 km is out-of-phase with that in temperature, consistent with the tropical MSAO. The lidar observations generally agree with satellite and meteor radar observations as well as

model simulations at similar latitude. The 50-70% of zonal momentum flux is induced by
short-period (10 min – 2 hr) GWs. The large zonal momentum flux in summer and winter due
to short-period GWs are clearly anti-correlated with eastward zonal wind maxima below 90
km, suggesting the filtering of short-period GWs by the SAO wind.

## 1. Introduction

The temperature and wind in the mesopause region (80-105 km) are key atmospheric parameters for studying the dynamics in this region. Ground-based instruments (e.g. lidars, radars), and space-borne instruments have been widely used to measure these key parameters over several decades (Vincent and Reid, 1983; She et al., 1998; Wu et al., 2008). Satellites can provide a near-global view of the mesopause region, but their local coverage is usually limited to two local times on the ascending and descending orbit. The lack of continuous coverage in local time makes it difficult to extract information on short period gravity wave (GW) perturbations from satellite data (Preusse et al., 2009). Ground-based meteor or medium frequency radars are capable of measuring mesopause wind in a continuous mode, but do not provide direct temperature measurements with sufficient accuracy and vertical resolution (Vincent and Reid, 1983). However, a narrowband sodium lidar is able to simultaneously measure mesopause region temperature and horizontal wind by utilizing the sodium high resolution spectrum (She et al., 1994; Arnold and She, 2003), which provides a unique opportunity to study GW perturbations and their breaking process in the mesopause region (Li et al., 2005; Li et al., 2007).

The long-term lidar observations have been used to study the seasonal variability of mesopause region temperature (She et al., 1998; Gardner et al., 2002; Xu et al., 2006; Friedman et al., 2007) and horizontal wind (Franke et al., 2005; Gardner et al., 2007), as well as sodium density (She et al., 2000; Gardner et al., 2005; Ejiri et al., 2010; Yi et al., 2009; Yuan et al., 2012), iron density (Yi et al., 2009; Lübken et al., 2011) and potassium density (Friedman et al., 2002; Plane et al., 2015). These datasets are extremely valuable to validate satellite results (Xu et al., 2006; Fan et al., 2007a; Dawkins et al., 2014) and improve general circulation models (Yuan et al., 2008; Feng et al., 2013; Marsh et al., 2013). When GWs break or dissipate in the mesopause region due to increased amplitudes or approaching critical level (where wave phase speed equal to horizontal background wind), they tend to deposit wave energy and momentum into the background flow, and further modify the temperature and wind near the breaking region (Lindzen et al., 1981; Liu and Hagan, 1998; Li et al., 2007). Therefore, measurements of the GW vertical flux of horizontal momentum and heat are critical for evaluating the GW contribution to the background state in this region, and their key roles in the dynamic coupling between lower and middle/upper atmosphere (Li et al.,

67  2013; 2016).

The vertical flux of horizontal momentum can be directly derived from the vertical wind
perturbation and associated horizontal wind perturbation. To ensure accuracy of the GW
momentum flux, the wind data must have high temporal and vertical resolutions with good
precision and a long-time average (Kudeki and Franke, 1998; Thorsen et al., 2000). Several
studies of lidar-observed GW momentum flux in the mesosphere/lower thermosphere (MLT)
region have been carried out previously (Espy et al., 2004; Gardner and Liu, 2007; Acott et al.,

2009).

In this paper, we present the seasonal variation of sodium density, temperature, zonal
wind and GW zonal momentum flux observed by the University of Science and Technology
of China (USTC) sodium temperature/wind lidar from January 2012 to December 2016 over
Hefei, China (32°N, 117°E). This is the first time simultaneous observations of the seasonal
variability of mesopause region temperature, zonal wind, and GW momentum flux by sodium
lidar over the Eastern Asia region have been reported. We compare the lidar results with
temperature observed by the Sounding of the Atmosphere Using Broadband Emission
Radiometry (SABER) instrument onboard the Thermosphere–Ionosphere–Mesosphere
Energetics and Dynamics (TIMED) satellite (Russell et al., 1999); zonal wind observed by a
nearby meteor radar (Xiong et al., 2004); and sodium density observed by the Optical
Spectrograph and InfraRed Imager System (OSIRIS) onboard the Odin satellite (Llewellyn et
al., 2004). These measurements are then compared with simulations from the Whole
Atmosphere Community Climate Model version 5 (WACCM) (Marsh et al., 2013; Mills et al.,
2016; Feng et al., 2017), using an updated meteoric input function (MIF) for Na
(Cárrillo-Sanchez et al., 2016). The instruments, datasets, and data analysis method are
described in section 2, followed by the results of temperature and sodium density in section 3,
and zonal wind and GW zonal momentum flux in section 4. A summary is provided in section

5.


**2. Instruments, datasets and analysis method**
The USTC sodium temperature/wind lidar, located on campus in Hefei, China (32°N,
117°E), utilizes a narrowband three-frequency design and can simultaneously observe sodium
density, zonal wind and temperature in the mesopause region during nighttime clear sky

conditions (Li et al., 2012). The system was initially set up in October 2011 with two receiving telescopes (30-inch diameter) pointing eastward and northward 30° from zenith for measuring the zonal and meridional wind, respectively. The output laser beam is split into two beams, each aligned parallel to one telescope. Between December 2012 and May 2014 (total 83 nights), the two receiving telescopes were pointed to eastward and westward, each 15° from zenith. This dual-beam setup allows us to derive the GW zonal momentum flux as well as the zonal wind. Since June 2014, the westward telescope was pointed to vertical for measuring the vertical fluxes of heat and sodium atoms, and the eastward telescope to 30° from zenith for measuring zonal wind. Between January 2012 and December 2016, we obtained 154 nights (~1400 hours) of valid data, which is sufficient to study the seasonal variations of sodium density, temperature, zonal wind, and GW momentum flux (83 nights) in the mesopause region over Hefei. Figure 1 shows the number of nights with valid datasets in each month of the different years. It is clear that Hefei has more clear nights in fall and winter than in spring and summer.

The Wuhan (31°N, 114°E) meteor radar, located at ~300 km west of Hefei, has measured mesopause region horizontal wind since January 2002 (Xiong et al., 2004). The vertical and temporal resolutions of radar wind are 3 km and 2 hr, respectively. The SABER instrument onboard the TIMED satellite can measure the near-global vertical profile of temperature from the lower stratosphere to the lower thermosphere (Russell et al., 1999). The SABER temperature dataset used in this paper is Level2A version 2.0, which has a vertical resolution of 2 km and accuracies of ±1-2 K between 75 and 95 km, increasing to ±4 K at 100 km. The OSIRIS instrument onboard the Odin satellite measures solar-pumped Na resonance fluorescence from a sun-synchronous polar orbit (Llewellyn et al., 2004), and the datasets can be used to retrieve the global vertical profiles of sodium density between 75 and 110 km with a ~10% uncertainty for 2 km vertical resolution (Gumbel et al., 2007; Fan et al., 2007a).

To compare with lidar results, we also use the temperature, zonal wind, and sodium density simulated by the WACCM, a chemistry-climate model which extends from the Earth's surface to the lower thermosphere (~140 km) (Garcia et al., 2007; Marsh et al., 2013a). WACCM uses the framework from the fully coupled global climate model Community Earth System Model (CESM version 1, e.g., Hurrell et al., 2013). In this paper, we use a version of WACCM described in Mills et al. (2016), which includes all the detailed physical processes

as described in the Community Atmosphere Model, version 5 (CAM5) (Neale et al., 2012).
The current configurations for WACCM are based on a finite volume dynamical core (Lin,
2004) for the tracer advection as well as a new surface topography data from Lauritzen et al.
(2015). WACCM has the fully interactive chemistry described in Mills et al. (2016), and we
have included the Na chemistry scheme listed in Plane et al. (2015) and Gomez Martin et al.
(2015, 2017), with an updated meteoric input function (MIF) for Na (Cárrillo-Sanchez et
al.,2016). The new MIF is calculated for the ablation of cosmic dust particles from Jupiter
Family Comets (80% by mass), Asteroids (8%), and Long Period Comets (12%), and the
injection rate of Na is about 8 times larger than that used in Marsh et al. (2013). The peak Na
ablation rate from Cárrillo-Sanchez et al. (2016) occurs around 87 km, which is ~15 km lower
than the MIF used in Marsh et al. (2013), which was based on meteor head radar
measurements which were biased to the high velocity dust particles which mostly originate
from Long Period Comets (Cárrillo-Sanchez et al., 2016). The absolute Na MIF used in this
paper has been divided by a factor of 5 from that in Cárrillo-Sanchez et al. (2016), in order to
match the observed Na layer density. This most likely reflects the fact that WACCM
underestimates the rate of vertical transport of Na species in the MLT because sub-grid
gravity waves are not resolved in the model (Huang et al., 2015). The horizontal resolution of
WACCM is 1.9˚ latitude by 2.5˚ longitude. The vertical model layers and the vertical
resolution are the same as Mills et al. (2017), which is 70 and ~3 km in the MLT region.
Although the model can be nudged by a re-analysis dataset, in the current study we have used
a "free-running" model simulation, which produces a satisfactory Na climatology in the
model. We ran the model for year 2000 condition for 13 years.

The lidar raw photon counts are first analyzed to generate hourly mean vertical profiles

of sodium density, temperature and line-of-sight (LOS) wind with 2 km vertical resolution for
each direction. Before and after the dual-beam setup (eastward-westward) between December
2012 and May 2014, we assume that the hourly mean vertical wind is negligible and then
derive the hourly mean zonal wind from the east channel LOS wind (eastward pointing at 30°
from zenith). During the dual-beam setup, we derive the hourly mean zonal wind profiles by
subtracting the hourly westward LOS wind from the eastward LOS wind and then dividing by
$2\sin\theta$ (e.g. $\theta$=20°) (Vincent and Reid,1983). The uncertainties of the hourly mean zonal wind
and temperature typically range from ~1.0 m/s and ~0.5 K at 92 km (Na peak layer) to ~6 m/s
and ~5 K at 82 km and 103 km (the edge of Na layer), respectively. We then generate the
nighttime hourly mean composite in each season.
Vincent and Reid (1983) presented a method utilizing the dual beam technique to derive
vertical flux of GW horizontal momentum, when two beams are pointed at equal and opposite
angle $\theta$ from the zenith. The zonal momentum flux $\overline{w'u'}$ is calculated as follows:
$$\overline{w'u'} = \frac{\overline{v^2(\theta,R)} - \overline{v^2(-\theta,R)}}{2\sin(2\theta)} \tag{1}$$
where $\overline{v^2(\theta,R)}$ and $\overline{v^2(-\theta,R)}$ are the square of the LOS wind perturbations in the east and
west channels respectively, and $\theta$ is the zenith angle (e.g. 20°). To derive the momentum flux,
we employed a similar procedure to that of Gardner and Liu (2007). Briefly, we first analyze
lidar raw photon counts to generate the LOS wind with a temporal resolution of 5 min and a
vertical resolution of 2 km. Data points with errors larger than 5 m/s were discarded during
the quality check. We remove the linear trend and nightly mean from the LOS wind to form
wind perturbations for each night. Data where the perturbation variances are smaller than the
corresponding noise variances are also excluded. The seasonal mean vertical profile of
perturbation variance is then obtained by averaging all available perturbation variances in that
season. This process is done separately for each beam. Finally, the seasonal mean momentum
flux is calculated using equation (1). In this way, the results only account for the GW
perturbations with periods of 10 min-20 hr and vertical wavelengths of 4-30 km. We also
apply a high-pass filter with cutoff at 2 hr on raw perturbations to examine the relative
contribution of short-period GWs (10 min – 2 hr) to total momentum flux.
Since the meteor radar observed zonal wind is only available in 2013, we then calculate
monthly mean with all available data for comparison. The SABER tracking points within ±5°
latitude band (27-37°N) and longitude band (112-122°E) of the lidar site are selected first. We
then discard the SABER temperature profiles that are outside of the lidar observation period.
Finally, we average all available SABER temperature profiles within each month to form the
monthly mean for comparison. A similar analysis method is used for the OSIRIS data. In the
case of WACCM, the zonal mean data are first extracted at the coordinates of the lidar site
and then the monthly mean profiles are generated in the same way as the lidar and radar
profiles.

## 3. Temperature and sodium density

Figure 2 shows the hourly mean temperature composite in four different seasons. The temperatures below 95 km are generally warmer in fall and winter than in spring and summer, consistent with the mesospheric residual meridional circulation with upwelling in the summer hemisphere and downwelling in the winter hemisphere (Andrew et al., 1987; Smith, 2012). It is most likely that the diurnal tide with downward phase progression dominates below 100 km in spring, although we only have 10-12 hr data. However, the tidal feature is not clear below 95 km in other seasons. The temperature above 100 km in all seasons clearly exhibits two minima after dusk and before dawn and a maximum near midnight, suggesting dominance and persistence of the semidiurnal tide in this latitude region throughout the year.

The clear downward phase progression of diurnal and semidiurnal tides in mesopause temperature was previously observed by sodium lidar at the Starfire Optical Range (SOR), New Mexico (35°N, 107°W) (Chu et al., 2005). However, their observations suggest a clear dominance of diurnal in April and October and semidiurnal in January below 100 km, while we see a clear dominance of diurnal only in spring (March-May), and mixed features in other seasons. In addition, the midnight maximum above 100 km shown in our results is not observed over SOR. The SABER observations reveal a diurnal amplitude of ~2 K and ~8 K, and semidiurnal amplitude of ~7 K and ~12 K at 95km for the USTC and SOR lidar sites, respectively (Zhang et al., 2010). This significant longitudinal variability is likely due to nonlinear interactions between the migrating tide and non-immigrating tide (Forbes et al., 2003) and stationary planetary wave number 1 (Lieberman et al., 1991), respectively, and/or tidal/gravity waves interactions (Lindzen, 1981; Liu and Hagan, 1998; Li et al., 2007; 2009). The clear longitudinal variability of tides between two lidar sites could thus cause significant differences in the nocturnal climatology.

Figure 3 shows the monthly mean of the nightly mean temperature observed by lidar and SABER, and simulated by WACCM. All three figures show qualitative agreement in the general pattern, but difference in absolute values. The mesopause is clearly located near 100 km in winter and below 95 km in summer, indicating a two-level mesopause as previously observed at mid- and high latitudes (von Zahn et al., 1996; She et al., 1998). The lidar observed temperature above 95 km is ~10 K lower than SABER, likely due either to the low signal-to-noise ratio in the lidar return signals above 100 km (Li et al., 2012), or to a non-local

thermal equilibrium influence in the SABER analysis (Mertens et al., 2001). The lidar
observed mesopause is also 5-10 K colder than that observed by SABER. The WACCM
simulated temperature is clearly higher than both sets of observations at most altitudes and
months. Yuan et al. (2008) showed a significant monthly mean mesopause region temperature
difference between lidar observations and WACCM simulations over Fort Collins, CO (41°N,
105°W); their comparisons show that the WACCM-simulated winter mesopause is much
warmer than measured by lidar, and the summer mesopause is ~3 km lower than lidar
observations. Another interesting feature in all three figures is that we see a temperature
maximum near ~90 km in March and April, and a second maximum in September and
October, likely related to the mesospheric semiannual oscillation (MSAO) usually dominant
in the equatorial middle atmosphere (Dunkerton, 1982; Burrage et al., 1996; Garcia et al.,

1997).

Our measured monthly means of the nightly mean temperatures are also generally
consistent with previously lidar observations at SOR (Gardner and Liu, 2007) and Fort
Collins, CO (She et al., 1998; Yuan et al., 2008). However, the SOR lidar observations were
~10 K colder below 90 km in summer, and ~10 K warmer between 90 and 95 km in spring,
suggesting significant differences between the two locations likely induced by the significant
longitudinal variability of the diurnal tide (Zhang et al., 2010). The semiannual oscillation
signature is evident over both Hefei and SOR between 90 and 95 km, but not over Fort
Collins. The summer mesopause observed by lidar over Hefei is clearly higher than over the
other two locations.
Figure 4 shows the hourly mean sodium density composite during the four different
seasons. The density increases with local time during the night, with a peak height around 92
km. The peak density is overall much higher in fall and winter than in spring and summer,
which is consistent with previous ground-based and satellite observations (She et al., 2000;
Fan et al., 2007a; Fussen et al., 2010). Some peaks above 95 km in summer are likely induced
by sporadic sodium layers (SSLs), which often occur in this season over Hefei (Dou et al.,
2010). The seasonal mean sodium peak density in winter can reach 4000-4500 $cm^{-3}$ after
midnight. Figure 5 shows the monthly mean of nightly mean sodium density observed by (a)
lidar and (b) Odin/OSIRIS, and simulated by (c) WACCM. Both observations agree well in
seasonal pattern and absolute sodium density, and are also consistent with the WACCM
model simulation. The elevated peak height and enhanced density in summer observed by
lidar is likely due to increased SSL events in summer over Hefei, which is neither frequently
observed by Odin/OSIRIS nor simulated by WACCM. The Odin/OSIRIS did observe SSLs
over China (Fan et al., 2007b), but probably less frequently at 0600 and 1800 local time than
at midnight. The observed sodium density over Hefei is quite consistent with previous
narrowband lidar observations over Fort Collins, CO (She et al., 2000) and Urbana, IL (States
and Gardner, 1998), but ~1.5 times higher than previous broadband sodium lidar observations
over the nearby city of Wuhan, China (Yi et al., 2009).
The variability of sodium density is clearly correlated with the temperature variability
shown in Figure 2. This is further demonstrated in Figure 6, where the correlation coefficient
between the composite temperature and relative sodium density perturbations is plotted using
lidar measurements (left) and the WACCM simulation (right). The temporal resolution for
both lidar and WACCM is 1 hr. We also examined the correlation in the four different
seasons and found no significant differences. The lidar observations are clearly consistent
with the WACCM simulation, and both results suggest a positive correlation with coefficient
of 0.5-0.8 between 80-90 km, but a negative correlation with coefficient of less than ~-0.4
above 96 km for lidar and 100 km for WACCM, consistent with lidar observations at Urbana
(40N) (Plane *et al*., 1999) and in the Arctic (Collins and Smith, 2004). However, our lidar
observations above 95 km are not consistent with the recent sodium lidar observations at
ALOMAR, which showed a positive correlation with temperature above this altitude (Dunker
et al., 2015). This difference may be related to energetic particle precipitation at high latitudes,
but the detailed mechanism is beyond scope of this paper.
Our lidar observations suggest that the main chemistry below 95 km is likely dominated
by neutral sodium chemistry, which essentially involves the partitioning of the metal between
atoms and the main reservoir $NaHCO_3$; the significant activation energy of the reaction
$NaHCO_3$ + H drives the balance towards Na at higher temperatures. In contrast, above 95 km
the source of atomic Na is from $Na^+$, which involves formation of cluster ions that then
undergo dissociative recombination with electrons; the formation of cluster ions is favored at
lower temperatures, hence the negative correlation coefficient between and Na and
temperature on the topside of the Na layer (Plane et al., 2015).

**4. Zonal wind and gravity wave momentum flux**

Figure 7 shows the hourly mean zonal wind composite in 4 different seasons. We see strong tidal oscillations with downward phase progression in all seasons, much clearer than those in temperature (Figure 2). The diurnal tide with vertical wavelength of ~ 20 km dominates in both spring and fall, while the semidiurnal tide with vertical wavelength of 30-40 km dominates in winter. In spring, the diurnal tide in temperature (Figure 2a) leads that in zonal wind by ~4hr between 90 and 95 km, consistent with earlier mid-latitude observations (Yuan et al., 2006). There is a strong wave oscillation signature with a period of ~8hr and amplitude of ~20 m/s that dominates in summer, possibly related to the terdiurnal tide. Previous observations by the nearby Wuhan meteor radar show that the diurnal amplitude near 90 km during equinox is ~ 30 m/s, with a semidiurnal amplitude of ~10 m/s (Xiong et al., 2004; Zhao et al., 2005). The comparable amplitude (~10 m/s) of diurnal and semidiurnal in winter is also revealed by these radar observations, with which our observations are generally consistent.

We show in Figure 8 the monthly mean of the nightly mean zonal wind observed by (a) lidar, (b) Wuhan meteor radar, and (c) simulated by WACCM. The radar observed zonal wind is only available in 2013 for comparison. The general pattern of the lidar observed zonal winds agrees well with the radar winds, but are 5-10 m/s stronger. This is likely due to the different vertical and temporal resolutions, signal-to-noise ratio, and the mesurement methods, as well as the different locations. The lidar results exhibit a semiannual variation near 90 km with minima in March and August/September, and one maximum in May/June, clearly out-of-phase with the temperature semiannual variation (Figure 3a). The lidar observed semiannual variation in both wind and temperature is consistent with the tropical MSAO previously observed by satellites (Garcia et al., 1997), and simulated by WACCM (Richter and Garcia, 2006). The lidar and radar observations agree with the WACCM simulation below 90 km in both pattern and magnitude, while disagreeing above. Interestingly, a recent comparision between lidar measurements over Fort Collins, CO and several general circulation models also reveals significant differences (Yuan et al., 2008).

The USTC lidar telescopes were pointed 15° from zenith in eastward and westward directions between December 2012 and May 2014. This setup allows us to derive the vertical flux of GW zonal momentum. A total of 83 nights of GW momentum flux measurements

were obtained with 21, 12, 23, and 27 nights in spring, summer, fall, and winter respectively. Figure 9 shows vertical profiles of the seasonal mean GW zonal momentum flux for period 10min – 16hr (blue) and 10min – 2hr (green), and zonal wind (red) in (a) spring, (b) summer, (c) fall, and (d) winter. The zonal momentum flux is mostly eastward in spring, positively correlated with the eastward zonal wind. However, the zonal momentum flux is mostly westward in other seasons, clearly anti-correlated with the eastward zonal wind, suggesting zonal wind filtering of GWs below 80 km. It is also clear that the zonal momentum flux induced by short-period (10 min – 2 hr) GWs clearly dominates total momentum flux in all seasons except summer.

The seasonal variation of zonal momentum flux is consistent with previous sodium lidar observation at SOR, NM (Gardner and Liu, 2007). However, MU radar observations near Kyoto, Japan (35°N, 136°E) shows a clear eastward flux in summer and westward flux in winter between 65 and 85 km (Tsuda et al., 1990). MF radar observations in Adelaide, Australia (35°S, 138°E) suggest an eastward flux of ~3 $m^2/s^2$ in winter (Reid and Vincent, 1987). We note here that part of the differences between our lidar results and other published work is likely due to different vertical and temporal resolutions and thus sensitivity to different portions of the GW spectrum. Table 1 compares the GW zonal momentum flux measured at different mid-latitude lidar and radar stations. The results from other locations are estimated from the following studies: Gardner and Liu (2007) for the SOR lidar results; Acott et al. (2009) for the Fort Collins, CO lidar results; and Tsuda et al. (1990) for the Japan MU radar results. This comparison demonstrates that all observations report a clear westward GW zonal momentum flux in winter. In spring, both the USTC and SOR lidars observed an eastward momentum flux of 1.4-2 $m^2/s^2$.

The short-period (10 min – 2 hr) GWs clearly contribute 50%-70% of the total momentum flux, consistent with previously medium frequency (MF) radar observations (Fritts and Vincent, 1987). The large westward momentum fluxes of -0.9 and -0.6 $m^2/s^2$ for short-period GWs in summer and winter respectively are clearly anti-correlated with eastward zonal wind maxima below 90 km (Figure 8a), suggesting the filtering of short-period GWs by the SAO wind. However, this SAO variation is not clear in the total momentum flux. For the annual mean, our lidar result is clearly smaller than the SOR lidar result, mainly due to significant difference in summer. Our results also show that the annual mean zonal wind

averaged between 87-95 km is ~10 m/s eastward, and anti-correlated with the westward momentum flux of ~-0.15 $m^2/s^2$ induced by short-period GWs. This anti-correlation suggests that the GW momentum flux observed in the mesopause region is generally consistent with the wind filtering theory proposed by Lindzen (1981), and adopted by general circulation models (e.g. Richter et al., 2010).

**5. Summary**

Between 2012 and 2016, the USTC sodium temperature/wind lidar observed mesopause region nighttime temperature, zonal wind, and sodium density over 150 nights, and the vertical flux of zonal momentum during 83 nights. The seasonal nighttime hourly composites of temperature and zonal wind show clear diurnal and/or semidiurnal tidal signatures. In temperature, the diurnal tide with clear downward phase progression dominates only in spring, while the semidiurnal tide dominates above 100 km throughout the year. In zonal wind, the diurnal tide with vertical wavelength of ~ 20 km dominates in both spring and fall, while the semidiurnal tide with vertical wavelength of 30-40 km dominates in winter. Between 90 and 95 km, the diurnal tide in temperature in spring leads that in zonal wind by ~4 hr, consistent with previous observations and model simulations. The monthly mean results show a signature of semiannual variation in both temperature and zonal wind near 90 km but with clear out-of-phase feature, consistent with the tropical MSAO. Comparison of the USTC lidar results with observations by satellite and meteor radar, and simulated by WACCM show generally good agreement, although there are some differences among them, with pronounced disagreement between the observed zonal wind and the model above 90 km.

The seasonal mean of zonal momentum flux is mostly westward in summer, fall and winter, clearly anti-correlated with the eastward zonal wind, which suggests zonal wind filtering of GWs below 80 km. However, during spring the zonal momentum flux is mostly eastward, positively correlated with the eastward zonal wind. The short-period GWs clearly contribute 50%-70% of total momentum flux averaged over 87-95 km. The large westward momentum fluxes in summer and winter for short-period GWs are clearly anti-correlated with eastward zonal wind maxima below 90 km (Figure 8a), suggesting the filtering of short-period GWs by the SAO wind. The annual mean flux averaged over 87-95 km is ~-0.15 $m^2/s^2$ (westward) induced by the short-period GWs, anti-correlated with the zonal wind of

~10 m/s (eastward), suggesting that the GW momentum flux observed in the mesopause region is generally consistent with the wind filtering theory.

The sodium density increases with local time during the night, with a peak height near 92 km. The peak density is overall much higher in fall and winter than in spring and summer. The seasonal mean sodium peak density in winter can reach 4000-4500 $cm^{-3}$ after mid-night. The variability of sodium density is positively correlated with temperature variability, suggesting that chemistry plays a dominant role in the formation of sodium atoms in the mesopause region below 95 km. The lidar observations agree well with Odin/OSRIS satellite observations in both seasonal pattern and absolute monthly mean sodium density, consistent with WACCM simulations using a new Na meteoric input function.

**Acknowledgments**

The work described in this paper was carried out at the University of Science and Technology of China (USTC), under support of the National Natural Science Foundation of China grant 41674149 and the Open Research Project of Large Research Infrastructures of CAS - "Study on the interaction between low/mid-latitude atmosphere and ionosphere based on the Chinese Meridian Project. WF and JMCP were supported by the European Research Council (project 291332-CODITA). The National Center for Atmospheric Research (NCAR) is sponsored by the National Science Foundation. We thank Chengyun Yang, Shengyang Gu, Xianyu Wang, Yetao Cen, Feng Li, and Huazhi Ge for help to take lidar data. TL would like to thank Alan Liu for helpful discussion. The SD-WACCM model was obtained from the NCAR and run at the University of Leeds and is available for contacting the co-authors FW or JMCP. We would like to thank Francis Vitt at NCAR for the WACCM model support. The SABER data is downloaded from http://saber.gats-inc.com/. We thank Richard Collins and another anonymous reviewer for their constructive comments.

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

**Table 1.** Comparison of the GW zonal momentum flux ($m^2/s^2$) measured at different middle
latitude lidar and radar stations.

| Stations | Altitude/filter | Annual | Spring | Summer | Fall | Winter |
|---|---|---|---|---|---|---|
| USTC lidar (32°N, 117°E) | 87 – 95 km 10min – 16hr | -0.08 | 1.4 | -0.2 | -0.3 | -0.9 |
| | 87 – 95 km 10min – 2hr | -0.15 | 0.8 | -0.9 | -0.16 | -0.6 |
| SOR lidar (35°N, 107°W) | 85 – 100 km 3min – 14hr | -1.2 | ~2 | 1.8 | N/A | -1.7 |
| CSU lidar (41°N, 105°W) | 85 – 95 km 6min – 4hr | N/A | ~0.1 | N/A | ~0.1 | -0.7 |
| MU Radar (35°N, 136°E) | 65 – 85km 5min – 2 hr | N/A | ~0 | 2.0 | ~0 | -1.5 |


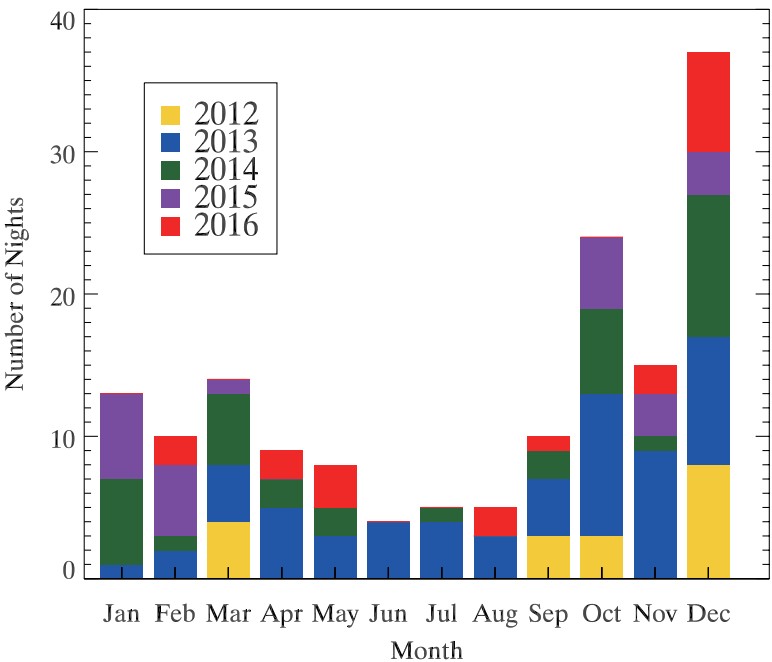


**Figure 1.** Histogram of number of nights with valid data observed by the USTC sodium lidar.

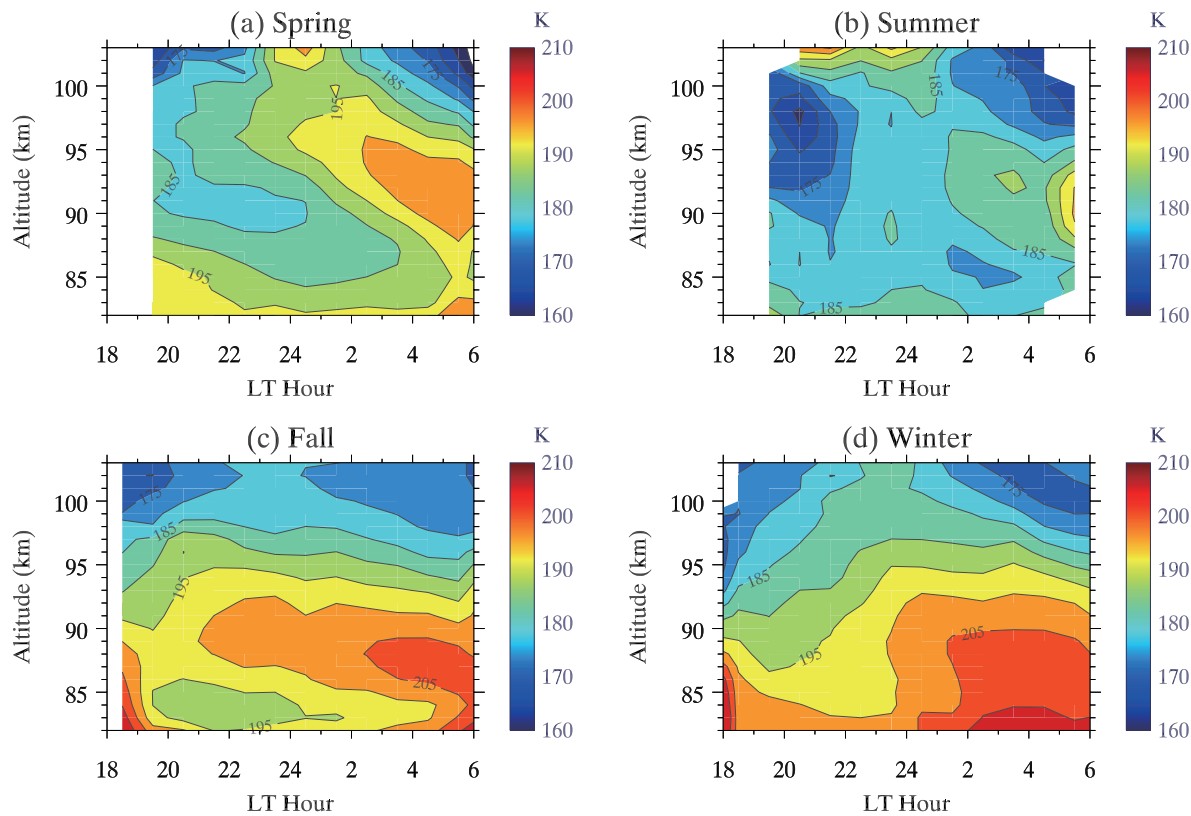

**Figure 2.** Lidar observed nighttime hourly mean temperature composite in (a) spring, (b) summer, (c) fall, and (d) winter.

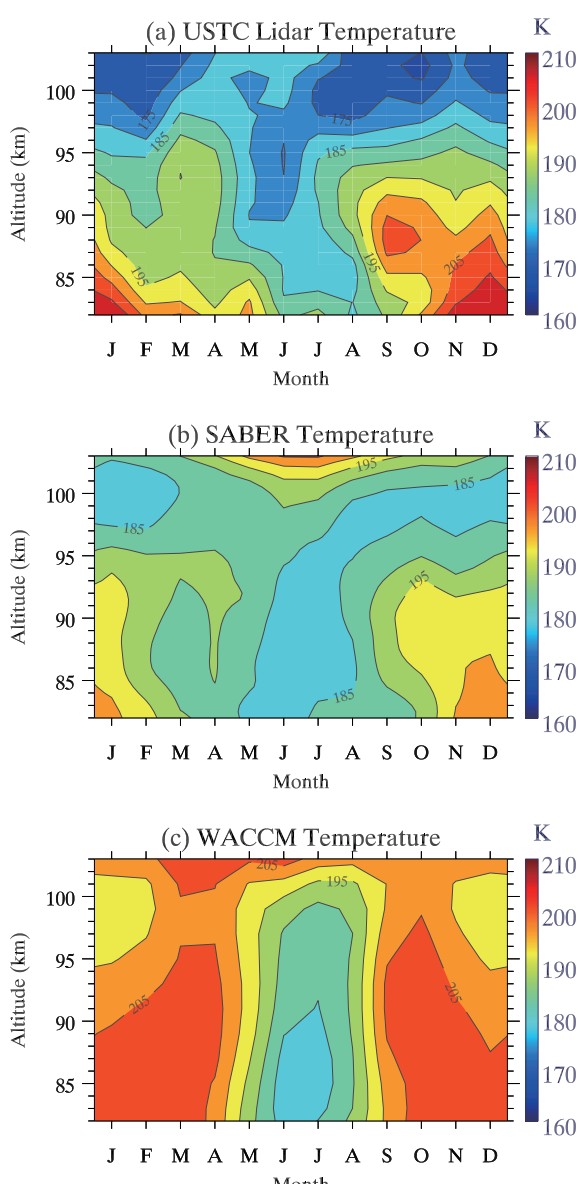


**Figure 3.** Monthly mean of mean temperature observed by (a) lidar, (b) SABER, and

simulated by (c) WACCM.

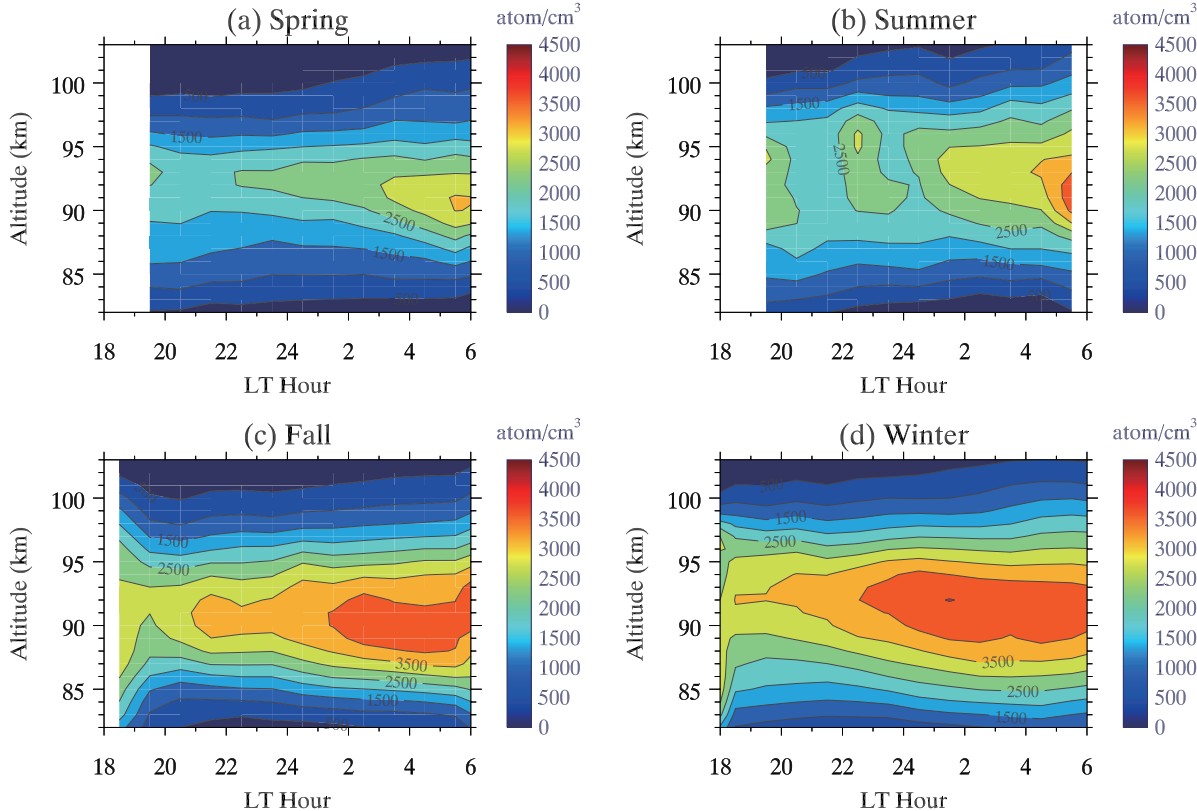


**Figure 4.** Same as Figure 2, but for sodium number density.

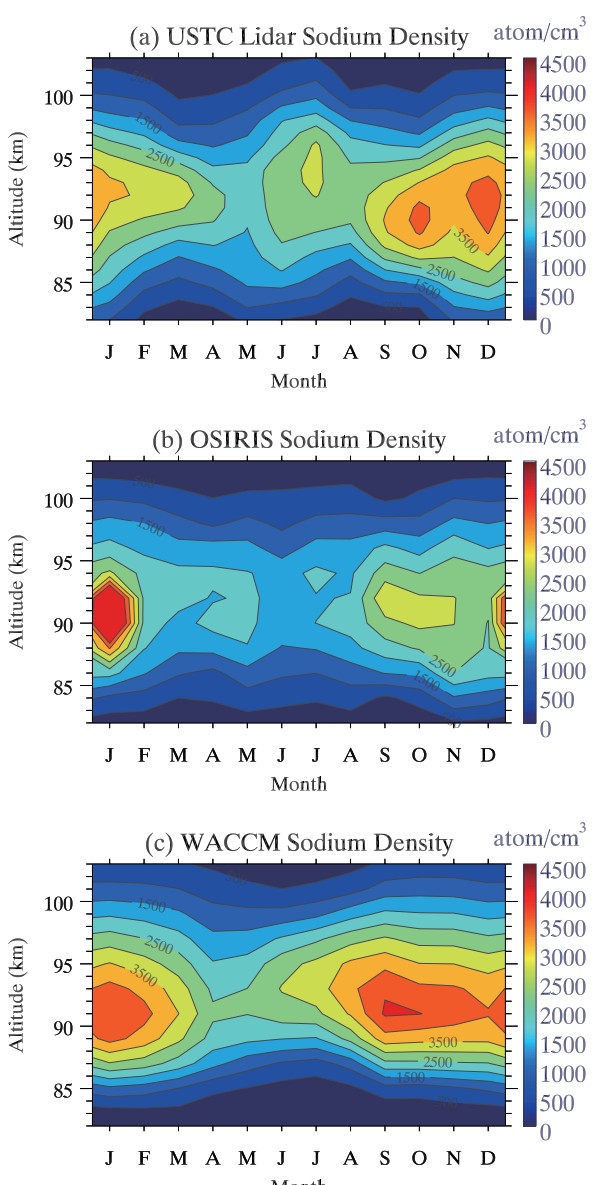


**Figure 5.** Monthly mean of nightly mean sodium density observed by (a) lidar and (b) Odin/

OSIRIS, and simulated by (c) WACCM.

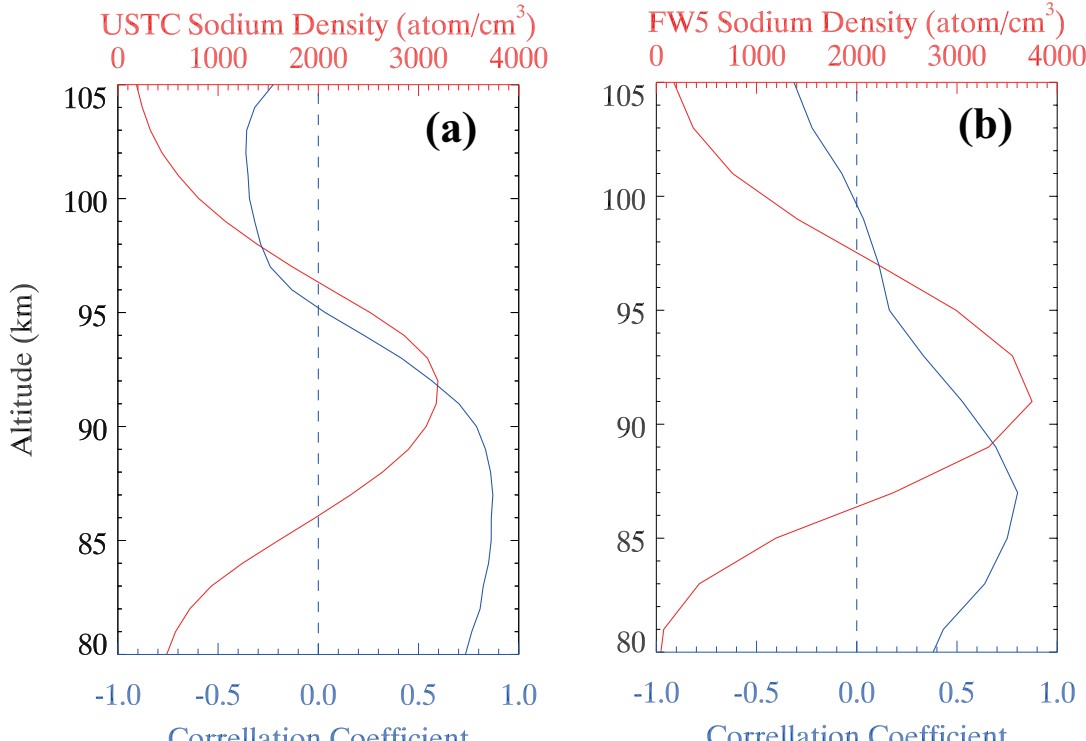

**Figure 6.** The vertical profiles of correlation coefficent (blue) between composite temperature and relative sodium density perturbations, and annual mean sodium density (red), observed by lidar (left) and simulated by WACCM (right).

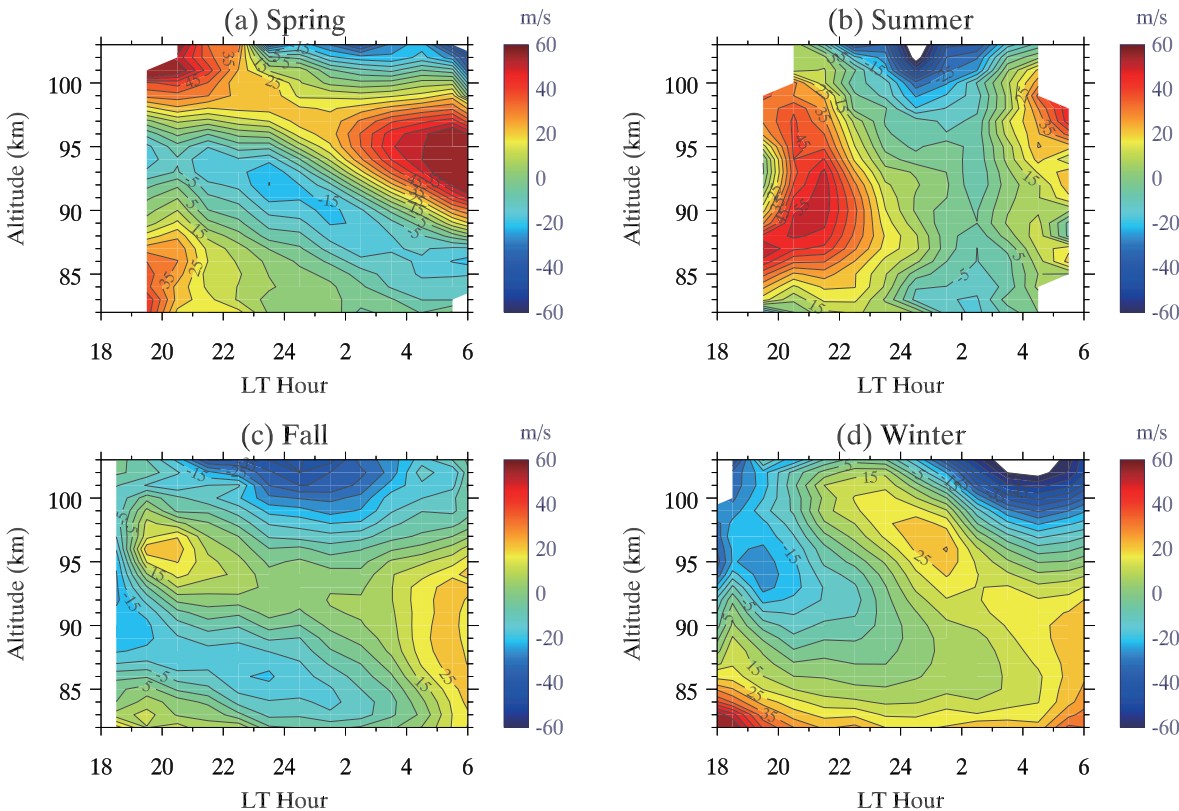

**Figure 7.** Same as Figure 2, but for zonal wind.

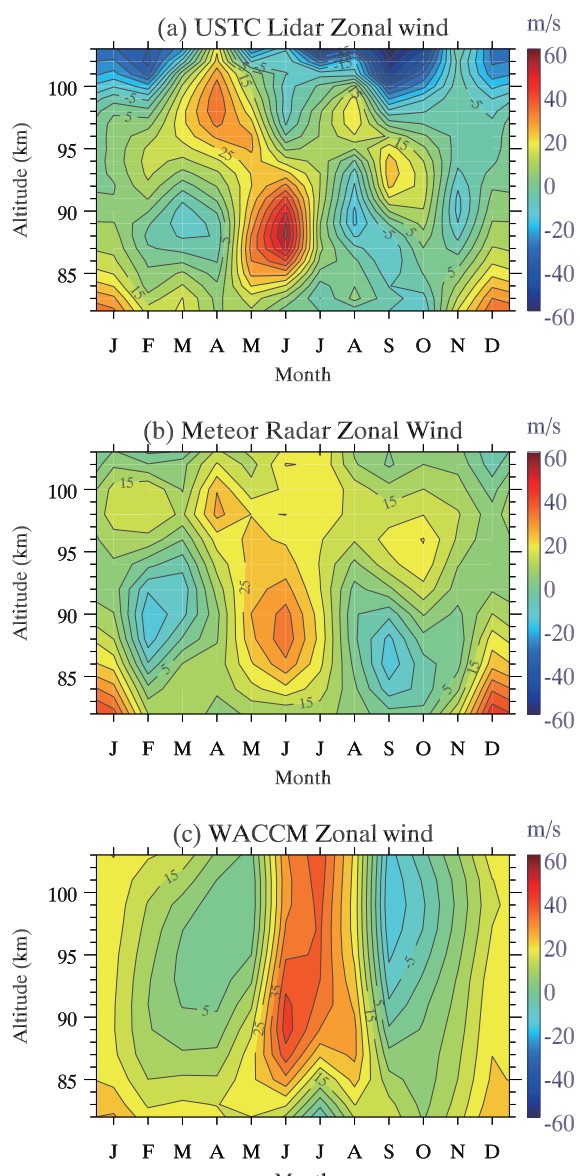

652

**Figure 8.** Monthly mean of nightly mean zonal wind observed by (a) lidar, (b) meteor radar,

and simulated by (c) WACCM.

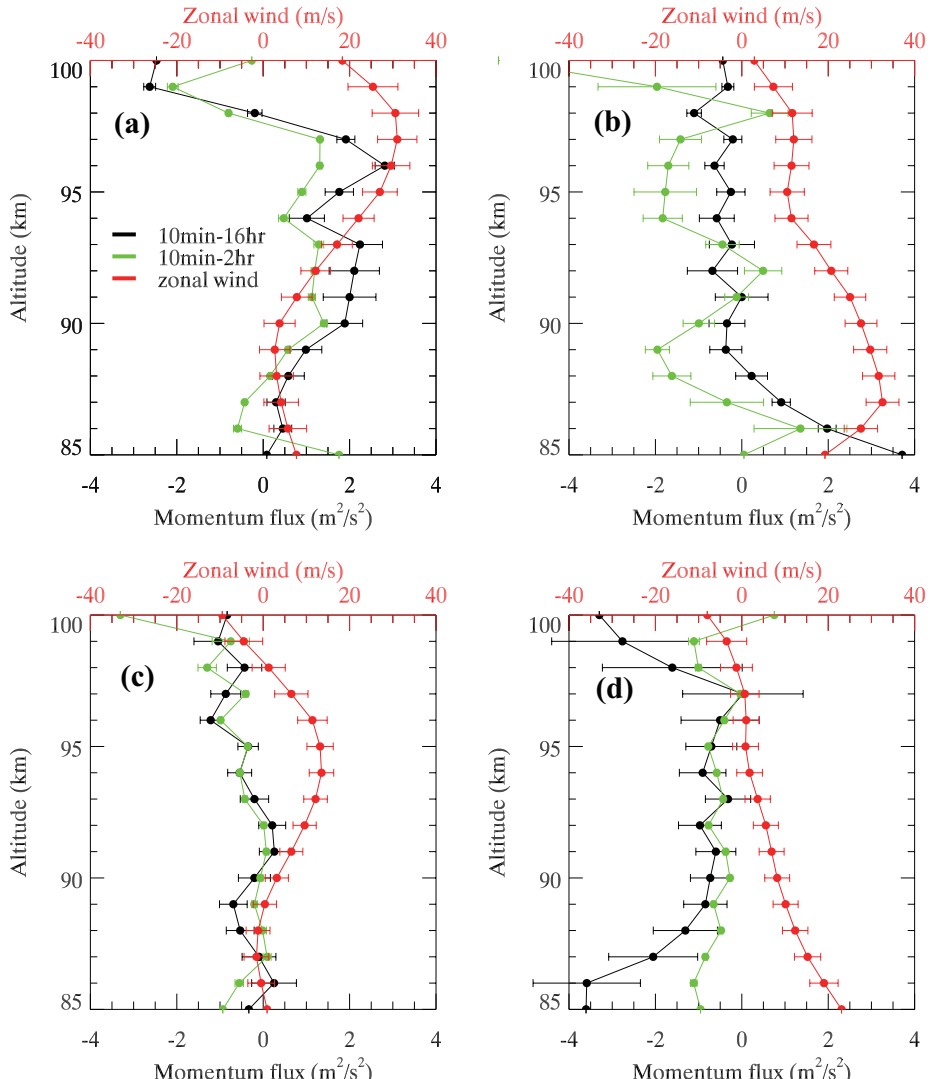

**Figure 9.** Comparision of seasonal mean of nightly mean zonal wind (red) and zonal momentum flux for 10min - 16hr (blue) and 10min - 2hr (green) observed by lidar in (a) spring, (b) summer, (c) fall, and (d) winter.