# Peer review of "Climatology of mesopause region nocturnal temperature, zonal wind, and sodium"

_Atmospheric Chemistry and Physics, 2018_

## Short Comment (SC1) · 4 May 2018

"Climatology of mesopause region nocturnal temperature, zonal wind, and sodium density observed by sodium lidar over Hefei, China (32° N, 117° E)" by Li et al.

General comments; This paper presents a new multi-year set of sodium wind-temperature lidar observations of the mesopause region from Heifei, China (32°N, 117°E). These observations over six years have yielded 237 nights (∼2200 hours) of mesospheric sodium, temperature, and zonal wind measurements that support the

investigation of chemistry and dynamics on time scales of hours over the night and months over the year. The observations are also compared with a contemporary coupled chemistry climate model and show reasonable agreement. The lidar system is a state-of-the-art wind-temperature system and the data set represents a significant experimental and observational effort.

Specific Comments; The seasonal and tidal dynamics presented in the paper is in reasonable agreement with other mid-latiude studies. The paper presents yearly plots of monthly mean temperature, zonal wind and sodium density. The paper highlights the presence of the tropical mesospheric semi-annual oscillation where the temperature is in quadrature with the zonal winds. The paper presents nightly composite plots of hourly temperature, zonal wind and sodium density for each season. The paper highlights the presence of semidiurnal and diurnal tides in the winds and temperatures based on visual inspection of the composites. The tidal signatures are more pronounced in the zonal wind than the temperature. The diurnal tide appears to dominate at the equinoxes, while the semidiurnal tide dominates in winter. There is no clear tidal signature present in the summer composite. The authors note that these observations are consistent with meteor radar observations at Wuhan and sodium wind-temperature lidar observations Fort Collins. The authors may wish to compare their observations with those from Adelaide, Australia (35°S, 138°E).

The paper presents the correlation of sodium density and temperature perturbations. What was the time resolution of the data used in the correlation? The presence of a positive correlation below 95 km and a negative correlation above 95 km has been shown in steady state analysis of the chemistry of the sodium layer [e.g., Collins and Smith, JASTP, 2004]. The question of the time-scale of the correlations is important in assessing vertical fluxes in sodium density that are used to quantify meteoric input into the atmosphere [e.g., Gardner et al., JGR, 2014] and turbulent eddy fluxes in the mesosphere [e.g., Guo et al., GRL, 2017]. The authors might examine the correlation in different seasons and shed further light on the chemical dynamics that underlie the

correlation.

The paper presents momentum fluxes based on the coplanar beam technique [Vincent and Reid, JAS, 1984]. Again the authors show consistency with other mid-latitude observations showing an eastward flux in summer and a clearer westward flux in winter. The fluxes include fluctuations with periods from 10 min to 16 h. Studies have shown the that waves with high intrinsic frequencies are most effective at transporting momentum in the middle atmosphere [Fritts and Vincent, JAS, 1987]. The authors might consider calculating the fluxes by band-limiting the data to the higher frequencies (e.g., 10 min-4 h) and seeing if these periods dominate the fluxes. The seasonality of the fluxes might also be examined in terms of the semi-annual oscillation.

---

## Referee Comment (RC1) · Anonymous Referee #2 · 7 May 2018

The paper reports a multi-year data set in mesopause region that is important for upper atmosphere studies. The work also demonstrates the great capabilities of the USTC Na lidar. For this paper, I think it is also important to illustrate the differences among the three mid-latitude sites and discuss the geophysical implications these differences reveal. It would be quite helpful if plots of the climatology from the other two mid-latitude sites could be added along side that of USTC, provided that they are available. I also hope the author could spend some extra ink on the discussion of this topic. For example, if the diurnal tide dominates the altitude range below 100 km and could ac-

[Figure]

count for the difference of the climatology among the three sites, there is a possibility that different diurnal tidal components dominate the midlatitude of east Asia and north America. The most recent tidal wave climatology is open to public (both GSWM and CTMT), should be an easy check. The author also mentions the differences between the lidar and the nearby radar wind measurements, and attribute them to different resolutions. The lidar data can, then, be processed using the radar resolution, and see how it affects these differences. For the momentum flux measurements, it is critical to process the lidar data with the same temporal and spatial resolution as those in the literature for comparisons, due to the sensitive of these results to the GW spectrum.

The presentation is very clean and figures are easy to understand. There seems to be a typo in line 164 of w'u' under the par, however. Please correct.

---

## Author Comment (AC1) · 2 Jul 2018

We thank the reviewers for their constructive comments and suggestions, which have helped us to significantly improve our manuscript. Following the reviewers' comments and suggestions, we revised the manuscript (please see the manuscript with changes tracked). Our responses to the reviewers' specific comments (blue text) are detailed below with page and line numbers referring to the manuscript with "track changes" turned off i.e., the clean version. (Note: the complete pdf is in acp-2018-290-supplement.pdf)

[Figure]

Reviewer #1 comments (&SC1): General comments; This paper presents a new multi-year set of sodium wind-temperature lidar observations of the mesopause region from Hefei, China (32N, 117E). These observations over six years have yielded 237 nights (âĹij2200 hours) of mesospheric sodium, temperature, and zonal wind measurements that support the investigation of chemistry and dynamics on time scales of hours over the night and months over the year. The observations are also compared with a contemporary coupled chemistry climate model and show reasonable agreement. The lidar system is a state-of-the-art wind-temperature system and the data set represents a significant experimental and observational effort.

Response: Thank you for your great comments. This will encourage us to continue our routine and long-term lidar observations.

Specific Comments; (1) The seasonal and tidal dynamics presented in the paper is in reasonable agreement with other mid-latitude studies. The paper presents yearly plots of monthly mean temperature, zonal wind and sodium density. The paper highlights the presence of the tropical mesospheric semiannual oscillation where the temperature is in quadrature with the zonal winds. The paper presents nightly composite plots of hourly temperature, zonal wind and sodium density for each season. The paper highlights the presence of semidiurnal and diurnal tides in the winds and temperatures based on visual inspection of the composites. The tidal signatures are more pronounced in the zonal wind than the temperature. The diurnal tide appears to dominate at the equinoxes, while the semidiurnal tide dominates in winter. There is no clear tidal signature present in the summer composite. The authors note that these observations are consistent with meteor radar observations at Wuhan and sodium wind-temperature lidar observations Fort Collins. The authors may wish to compare their observations with those from Adelaide, Australia (35S, 138E).

Response: Following your suggestion, we have included a few sentences to compare our momentum flux with MF radar observation at Adelaide, Australia (a conjugate site). We did not compare the tidal signature with those determined from MF observations

at Adelaide, Australia, since we could only visually examine the tidal components from the data due to limited local time coverage, and so this is very preliminary.

Changes: Please see page 12 lines 326-328. "The MF radar observations in Adelaide, Australia (35°S, 138°E) suggest an eastward flux of $\sim$3 m2/s2 in winter (Reid and Vincent, 1987)."

(2) The paper presents the correlation of sodium density and temperature perturbations. What was the time resolution of the data used in the correlation? The presence of a positive correlation below 95 km and a negative correlation above 95 km has been shown in steady state analysis of the chemistry of the sodium layer [e.g., Collins and Smith, JASTP, 2004]. The question of the time-scale of the correlations is important in assessing vertical fluxes in sodium density that are used to quantify meteoric input into the atmosphere [e.g., Gardner et al., JGR, 2014] and turbulent eddy fluxes in the mesosphere [e.g., Guo et al., GRL, 2017]. The authors might examine the correlation in different seasons and shed further light on the chemical dynamics that underlie the correlation.

Response: The temporal resolution is 1hr for the vertical profiles of both sodium and temperature. We have included in the text both the suggested reference Collins and Smith (2004), as well as the original reference to this making these Na-T correlations (Plane et al., (1999)). We then examined the correlation in the four different seasons, but the differences are not significant as shown in the following figures (figure1, response_f1.png). We would therefore prefer to keep the existing plots.

Changes: Please see page 10 lines 263-265. "The temporal resolution for both lidar and WACCM is 1hr. We also examined the correlation in the four different seasons, but did not find a significant difference." Please see page 10 lines 268-269. "..., consistent with lidar observations at Urbana (40N) (Plane et al., 1999) and in the Arctic (Collins and Smith, 2004).."

(3) The paper presents momentum fluxes based on the coplanar beam technique [Vincent and Reid, JAS, 1983]. Again the authors show consistency with other mid-latitude observations showing an eastward flux in summer and a clearer westward flux in winter. The fluxes include fluctuations with periods from 10 min to 16 h. Studies have shown the that waves with high intrinsic frequencies are most effective at transporting momentum in the middle atmosphere [Fritts and Vincent, JAS, 1987]. The authors might consider calculating the fluxes by band-limiting the data to the higher frequencies (e.g., 10 min-4 h) and seeing if these periods dominate the fluxes. The seasonality of the fluxes might also be examined in terms of the semi-annual oscillation.

Response: Thank you for this important suggestion. We have recalculated the momentum flux with more careful handling of large value perturbations. The new high-pass filtered data with 10min-2hr were used to calculate the momentum flux, and we found that the momentum flux for GWs with period 10min-2hr accounts for 50%-70% of those derived from GWs with period 10 min-16hr. This suggests that the short-period GWs contribute to the majority of total momentum flux, consistent with early MF radar observation at Southern Hemisphere Adelaide, Australia (Fritts and Vincent, 1987). We also find that the large short-period GW momentum flux in summer and winter are clearly anti-correlated with eastward zonal wind maxima below 90 km, suggesting the filtering of short-period GWs by the SAO wind.

Changes: Please see page 12 lines 337-342. "The short-period (10min – 2hr) GWs clearly contribute 50%-70% of total momentum flux, consistent with previously medium frequency (MF) radar observations (Fritts and Vincent, 1987). The large westward momenta of -0.9 and -0.6 m2/s2 in summer and winter, respectively, for short-period GWs observed by our lidar are clearly anti-correlated with eastward zonal wind maxima below 90 km (Figure 8a), suggesting the filtering of short-period GWs by the SAO wind. However, this SAO variation is not clear in the total momentum flux."

Reviewer #2 Comments:

(1) The paper reports a multi-year data set in mesopause region that is important for

upper atmosphere studies. The work also demonstrates the great capabilities of the USTC Na lidar. For this paper, I think it is also important to illustrate the differences among the three mid-latitude sites and discuss the geophysical implications these differences reveal. It would be quite helpful if plots of the climatology from the other two mid-latitude sites could be added along side that of USTC, provided that they are available.

Response: Although we think this is a good suggestion, it would require a significant enlargement of the scope of the current study which focuses in detail on the Hefei results, and where we already compared with the published results from the two US lidar sites. A more detailed comparison of tides and gravity waves at the three sites could be the basis for a future study, which would involve contacting the PIs and gaining permission to use their data..

(2) I also hope the author could spend some extra ink on the discussion of this topic. For example, if the diurnal tide dominates the altitude range below 100 km and could account for the difference of the climatology among the three sites, there is a possibility that different diurnal tidal components dominate the mid-latitude of East Asia and North America. The most recent tidal wave climatology is open to public (both GSWM and CTMT), should be an easy check.

Response: Following your suggestion, we have included more discussion on the tides observed by SABER, and the contribution of tidal longitudinal variability to the observed differences in the nighttime climatology between the USTC and SOR lidars.

Changes: Please see page 8 lines 206-213. "The SABER observations reveal a diurnal amplitude of ∼2 K and ∼8 K, and semidiurnal amplitude of ∼7 K and ∼12 K at 95km for the USTC and SOR lidar sites, respectively (Zhang et al., 2010). This significant longitudinal variability is likely due to nonlinear interactions between the migrating tide and non-migrating tide (Forbes et al., 2003) and stationary planetary wave number 1 (Lieberman et al., 1991), and/or tidal/gravity waves interactions (Lindzen, 1981; Liu

and Hagan, 1998; Li et al., 2007; 2009). The clear longitudinal variability of tides between the two lidar sites could thus cause significant differences in the nocturnal climatology." Please see page 9 lines 235-238. "However, the SOR lidar observations were ∼10 K colder below 90 km in summer, and ∼10 K warmer between 90 and 95 km in spring, suggesting significant differences between the two locations likely induced by the significant longitudinal variability of the diurnal tide (Zhang et al., 2010)."

(3) The author also mentions the differences between the lidar and the nearby radar wind measurements, and attribute them to different resolutions. The lidar data can, then, be processed using the radar resolution, and see how it affects these differences.

Response: In the following figure (figure2, response_f2.png) we plot the lidar data with different temporal and vertical resolutions, compared with meteor radar results over two days. The left panel is for the lidar zonal wind with 2 km and 1 hr resolution, the middle panel is for 3 km and 2 hr resolution, and the right panel is for the radar wind with 3 km and 2 hr resolution. It turns out there is some difference. The main feature between 80-100 km agrees very well between lidar and radar, except the maximum values are somewhat different. But above 100 km there are significant differences, likely due to the low signal-to-noise ratio of both instruments. In the revised paper we have kept a 2 km and 1 hr resolution in figure 8, since we want to emphasize the higher vertical and temporal resolution of the lidar compared with the meteor radar.

Changes: Please see page 11 lines 300-302. "This is likely due to the different vertical and temporal resolutions, signal-to-noise ratio, and the measurement methods, as well as the different locations."

(4) For the momentum flux measurements, it is critical to process the lidar data with the same temporal and spatial resolution as those in the literature for comparisons, due to the sensitive of these results to the GW spectrum.

Response: Yes, we agree with you. However, due to the different signal-to-noise between our lidar and other instruments, we analyze the data with particular resolutions

(2 km, 5 min) which are optimal for deriving the momentum flux. Yes, part of the difference between our lidar results and previously published results is likely due to different vertical and temporal resolutions. We now include a sentence in the text to point this out. We also apply a high-pass filter with cutoff at 2hr to raw perturbations to examine the relative contribution of short-period GWs (10min - 2hr) to the total momentum flux. The short-period GWs contribute 50-70% of total momentum flux. The large short-period GW momentum fluxes in summer and winter are clearly anti-correlated with the eastward zonal wind maxima below 90 km, suggesting the filtering of short-period GWs by the SAO wind.

Changes: Please see page 12 lines 328-330. "We note here that part of the difference between our lidar results with previously published work is likely due to different vertical and temporal resolutions, and thus sensitivity to different portions of the GW spectrum". Please see page 12 lines 337-342. "The short-period (10 min – 2 hr) GWs clearly contribute 50%-70% of the total momentum flux, consistent with previously medium frequency (MF) radar observations (Fritts and Vincent, 1987). The large westward momentums of -0.9 and -0.6 m2/s2 in summer and winter, respectively, for short-period GWs observed by our lidar are clearly anti-correlated with eastward zonal wind maxima below 90 km (Figure 8a), suggesting the filtering of short-period GWs by the SAO wind. However, this SAO variation is not clear in the total momentum flux."

The presentation is very clean and figures are easy to understand. There seems to be a typo in line 164 of w'u' under the par, however. Please correct.

Response: Thanks. We have corrected it.

Please also note the supplement to this comment:
https://www.atmos-chem-phys-discuss.net/acp-2018-290/acp-2018-290-AC1-supplement.pdf

———————————————

[Figure]

**Fig. 1.**

**13365**

**13363**

**Fig. 2.**